# The Method of Soundscape Naturalness Curves in the Evaluation of Mountain Trails of Diversified Anthropopressure—Case Study of Korona Beskidów Polskich

Magdalena Malec *, Renata Kędzior ● and Agnieszka Ziernicka-Wojtaszek ●

Department of Ecology, Climatology and Air Protection, Faculty of Environmental Engineering and Land Surveying, University of Agriculture in Kraków, 30-059 Kraków, Poland
* Correspondence: magdalena.malec@urk.edu.pl; Tel.: +48-12-662-40-15 or +48-12-662-40-16

**Abstract:** Sound in the landscape is an element of the multisensory experience of the environment. In areas that are naturally valuable and additionally used for tourism, the quality of this element is much more important than in urban areas. The aim of the study was to assess the soundscape diversity of mountain trails included in the Crown of the Polish Beskids (Korona Beskidów Polskich). Two methods were used in the study: The first was sound intensity measurement using a sonometer, which provided information on the physical aspect of the landscape. The second method involved recording all sounds divided into two basic categories: anthropogenic and natural. These results made it possible to propose a new method for assessing the naturalness of the soundscape by plotting naturalness curves. In contrast to frequently used survey-based methods, in this method we minimise subjectivity, which is mainly due to the different perceptions of sounds by the assessors. Given how many psychophysical aspects can affect the reception and perception of sounds, the method of naturalness curves allows for a universal assessment of landscape quality. On all the mountain trails surveyed, the average sound intensity values exceeded 40 dB, which the authors considered to be borderline for areas of natural value and recreational use. In the study area, the influence of anthropopression on soundscape formation was found to be diverse and dependent on many factors. However, there was no clear evidence that tourism was the main negative influence. The plotted naturalness curves showed a large variation between trails, but not all trails showed a correlation between this parameter and the number of tourists on the trail.

**Keywords:** sustainable tourism; soundscape; mountain trails; naturalness curves

## 1. Introduction

In the middle of the 19th century, the first marked hiking trails were established in Poland, which was related to the increasing popularity of hiking and skiing. The first trails were marked in the Tatra Mountains by the Tatra Society. Nowadays it is difficult to imagine tourism, especially in mountainous areas, without specially designated, marked, and often also developed trails. Although the beginnings of qualified tourism in Europe date back to the early to mid-19th century, the system and network of hiking trails is very diverse. Nowadays, Poland and parts of central Europe have a very dense, and evenly distributed trail network, while in other European countries it is small or located only in selected regions [1]. Tourist attractiveness is usually assessed on the basis of features such as transport accessibility, tourist development, or tourist assets. Precisely among tourist qualities, landscape qualities, e.g., the presence of viewpoints or open panoramas, are one of the important elements [2–4]. Soundscape as an element of multisensory perception of the surrounding environment is an area neglected in the literature on tourism space [5–7]. As evidenced by some studies in many countries, tourists visiting especially national parks prefer the sounds of nature [8–11].

Sound as an element of the multisensory perception of the mountain landscape is, on the one hand, an important element of place identity and tourist attractiveness in both natural and cultural terms. On the other hand, tourist traffic, which often exceeds the absorption tests, poses a threat to its naturalness. In addition, the noise associated with it, e.g., traffic noise, not only masks the sounds characteristic of the region but poses a threat to nature [12,13]. Lynch et al. [14] and Barber et al. [15] indicate that noise can cause environmental fragmentation and general ecological stress. In addition, a number of authors highlight the negative impacts of tourism-related noise on wildlife behaviour and abundance [16–25].

Sound, especially the sounds of nature, is an important and intrinsic element of the landscape that further emphasises the identity of a place. When considering the relationship between tourism and the soundscape, the focus should be, on the one hand, on the potential for tourism to have a negative impact on increasing noise. On the other hand, one should look at sound as a potential for the development of sustainable tourism. In both of these views, we cannot consider the soundscape and its possible disturbance solely in terms of its physical characteristics, as is the case of noise protection measures. It is crucial to look at the soundscape in a holistic way, in line with the philosophy of the creator of this concept, Schafer [26], where in addition to the physically defined sound level, the source of these sounds is also very important. This is especially true for naturally valuable areas, where humans are not only the recipients of sound, but also its creator. As noted by many authors, noise reduction, on which the efforts of space managers are mainly focused, is an insufficient measure [27–30]. The quality of a soundscape is also influenced by the source of the sound—preferred especially by visitors in places such as national parks are the sounds of nature (the sound of leaves, sounds associated with water, birdsong) [7,12,31–33]. An additional element is the perception of a particular sound, as highlighted by Aletta et al. [30]—some sounds are perceived as pleasant and others as unpleasant regardless of the sound level. Sound can be an important element of a local tourism product, as nowadays the choice of a tourist destination is not only influenced by accommodations, culture, entertainment, or sports facilities, but also by the quality of the landscape in terms of sound [10,34,35].

The aesthetic assessment of the landscape, both visual and sound, is related to the process of perception, i.e., the conscious and subconscious reception, but also the comparison of all elements. Due to the fact that these processes take place in the human mind and are therefore associated with many conditions such as age, origin, sex, education, previous experience, emotional state, health, and many others, such an assessment is highly subjective. Among other things, the methods proposed in ISO 12913 [36–39] are characterized by high subjectivity—the assessment of whether a given sound is pleasant or unpleasant is highly individual. This may mean that the obtained results cannot be compared in different communities or even age groups. Aletta et al. (2019) confirmed that methods A and B proposed in ISO 12913 give similar results, but there are some differences. In addition, attention is drawn to the problems of adapting translations and vocabulary used in different countries and the resulting divergence of meanings [40,41].

Hence, objective methods are sought to assess the soundscape with a particular focus on the source of sound, which is of great importance in tourist areas used for recreation, where sounds of nature are sought that have a positive impact on the perception of the environment but also on human health and well-being. The aim of the study was to investigate the diversity of the soundscape of the nine mountain ranges of the Crown of the Polish Beskids. The study posed the following exploratory questions: (i) whether anthropopressure has an impact on the differentiation of the soundscape of the studied ranges used for tourism, and whether tourism itself is the main element of it; (ii) whether there are differences in the naturalness of the soundscape depending on time (season), and the studied tourist trail. A new element of the work is the plotting of soundscape naturalness curves for each mountain range studied. Based on these, it is possible to develop new methods of managing high-value soundscape trails in order to protect them.

## 2. Materials and Methods

### 2.1. Description of the Area of Study

The research was carried out on 9 selected hiking trails leading to the highest peaks of the Beskids, all or part of which are located within Polish borders. In 2002, which was declared the "International Year of Mountains" by the United Nations, the Bochnia branch of PTTK (the Polish Tourist and Sightseeing Society) proposed the establishment of a special branch badge "Crown of the Polish Beskids", which included the trails studied [42]. Such activities were aimed at getting tourists interested in the less known and popular mountain ranges, while at the same time reducing tourist traffic in the more heavily frequented regions such as Pieniny, Tatry, and Sudety.

According to the physical–geographical regionalisation developed by Kondracki [43] and its update [44], Pieniny belongs to the Central Western Carpathian subprovince and the macro-region of Obniżenie Orawsko-Podhalańskie. On the other hand, the mountain ranges selected for the study belong to two subprovinces: Outer Western Carpathians and Outer Eastern Carpathians, or Eastern Beskids.

In the case of Beskid Niski, Bieszczady (in this case Bieszczady Zachodnie, which is part of the whole Bieszczady Mountains in the territory of Poland, Ukraine, and Slovakia), the highest peaks within Polish borders were selected for the study (Figure 1.).

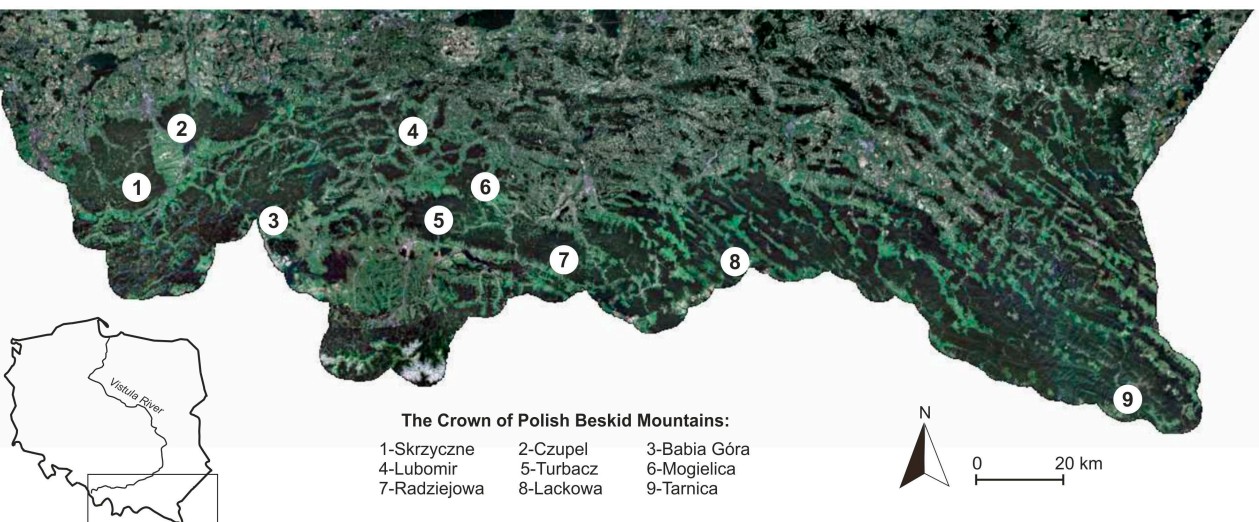

**Figure 1.** The localization of the studied Polish Beskid Mountains peaks.

Table 1 summarises all the peaks surveyed with their height and geophysical classification, as well as their starting point and number of measurement points.

### 2.2. Methods

Two groups of research methods were used to characterise and evaluate the soundscapes of the 9 mountain trails: objective and subjective. According to Liu and Kang [45], there are 5 subjective categories of soundscapes: definition, memory, sentiment, expectation, and landscape aesthetics. Human perception of the soundscape and its assessment are largely more important than the physical parameters of sound that can be measured objectively. The irritation caused by the impact of noise on the human sensory apparatus is only 30% dependent on the physical aspects of sound [46]. The soundscape is different from the acoustic environment, as it refers to perceptual constructs and not just physical phenomena.

**Table 1.** Summary of trails included in the study.

| Region | Peak | Height ASL (m) | Starting Location | Length of Trail (km) | Number of Measurement Points |
|---|---|---|---|---|---|
| **Beskid Żywiecki** | Babia Góra (Diablak) (BG) | 1725 | Zawoja Markowa | 6.30 | 17 |
| **Beskid Makowski** | Lubomir (LU) | 904 | Węglówka (Jaworzyce Pass) | 3.10 | 8 |
| **Beskid Mały** | Czupel (CZ) | 930 | Wilkowice | 6.85 | 15 |
| **Beskid Niski** | Lackowa (LA) | 997 | Wysowa Zdrój | 8.40 | 19 |
| **Beskid Wyspowy** | Mogielica (MO) | 1170 | Chyszówki (Rydza Śmigłego Pass) | 3.70 | 9 |
| **Beskid Sądecki** | Radziejowa (RA) | 1262 | Piwniczna Zdrój | 10.50 | 23 |
| **Beskid Śląski** | Skrzyczne (SK) | 1257 | Buczkowice | 7.01 | 15 |
| **Bieszczady Zachodnie** | Tarnica (TA | 1346 | Ustrzyki Górne | 8.20 | 18 |
| **Gorce** | Turbacz (TU) | 1315 | Rabka Zdrój | 15.77 | 34 |

The study was carried out during the summer of 2021 and winter of 2022. This made it possible to analyse the variability of the parameters studied in terms of the seasons, and thus the different intensity of tourist traffic, and the possible differences in the types of acoustic events recorded. On each of the surveyed hiking trails, measurements were taken at approximately 500 m sections (plus additional measurements, e.g., inside a shelter). Due to the different lengths of the trails, the number of measurement points varied from 8 to 34 (Table 1). As a subjective method for assessing the quality of the soundscape of mountain hiking trails, sound intensity measurements were used with a sonometer. The study used a digital decibel meter (correction characteristic A (dB) taking into account the sensitivity of the human sense of hearing and the time constant FAST) placed on a tripod 1.5 m above ground level. Due to the necessity of preserving all elements of the soundscape, such as the wind, which constitutes the "genius loci" of mountainous areas, measurements were made regardless of weather conditions [35,47,48]. This is a deviation from the indications as to the conditions for measurements in studies on, e.g., traffic noise, proposed, among others, by Bohatkiewicz [49]. According to them, one of the limiting factors is wind speed exceeding $5 \text{ m} \cdot \text{s}^{-1}$. A total of 21,107 single measurements was made in both seasons using the sampling method (summer: 10,774, winter: 10,333). A single measurement was made every 1 s, and the results were automatically stored in the device's memory.

The soundscape is rarely considered in terms of tourism and recreational land management. Therefore, as a second method, an analysis of all so-called soundscape events occurring was used. We can divide them into two main categories, natural and anthropogenic, and each of them additionally into several lower-order categories [35,47,48,50–52].

These were later used in an attempt to create a more objective method to determine the degree of naturalness of the soundscape. In the case of the soundscape, as with the visually perceived landscape, it is assumed that the most conducive to recreation is that which is natural in nature. Following this line of reasoning, the authors decided to assess the naturalness of the soundscape of tourist trails leading to the peaks included in the so-called Crown of the Polish Beskids. The assessment was carried out in terms of its naturalness, i.e., the number of recorded sound events included in this category. Of course, the subjective perception of sound sources should be taken into account, as not every sound originating from the natural world will be perceived by everyone as pleasant (e.g., the sound of thunder for some persons). Conversely, not every anthropogenic sound (e.g., music) will be perceived as negative. A similar subjectivity characterises many landscape valorisation methods, e.g., Wejchert's impression curve [53–58], on which the authors based their work.

In the naturalness curve of the soundscape, we are somewhat more objective. This is due to the lack of an emotional approach of the observer to individual acoustic events. What is assessed here was not the subjective pleasure of a particular sound, but whether it belongs to the category of natural or anthropogenic sounds. As with the typology of landscapes into natural or cultural, the contribution of natural and anthropogenic elements was assessed without the subjective evaluation of these elements.

The method consists of recording and identifying all acoustic events (heard and additionally recorded) and assigning them to the relevant natural or anthropogenic groups. The observations were made along a set trail at equal distances (500 m). It was not necessary to draw the curve separately for the left and the right side. A scale of 0 to 10 points was adopted to draw the curve, each point being the percentage of natural events in the total number of sounds recorded (Table 2).

**Table 2.** Criteria for assessing the naturalness of the soundscape.

| Scale | 0 | 1 | 2 | 3 | 4 | 5 | 6 | 7 | 8 | 9 | 10 |
|---|---|---|---|---|---|---|---|---|---|---|---|
| **Percentage of acoustic events (%)** | 0 | 1–10 | 11–20 | 21–30 | 31–40 | 41–50 | 51–60 | 61–70 | 71–80 | 81–90 | 91–100 |

Source: own work based on Wejchert [53,54].

As in Wejchert's impression curve method, on which the authors based their work, two limiting values of 3 and 7 points were adopted [53,54,58]. Below 3 points, the study area is characterised by an unnatural soundscape; such a site requires measures to reduce anthropogenic influence. Between 3 and 7 points, the naturalness of the soundscape is assessed as medium, and it also requires action, but to a low degree. Above 7, on the other hand, it is an area of exceptional soundscape value and does not require active protection, but actions to preserve it. In addition, all persons on the trails at the time of the survey were counted.

The distribution of the analysed dependent variables (sound level and values of the soundscape naturalness valence points) was checked using the Shapiro–Wilk test. Due to the lack of a normal distribution, the relationship between the type of trail/mountain range and the parameters tested was determined using a generalised linear model (GLM) for Poisson distributions, where the connecting function was a logarithmic function. Statistical analyses were performed using Statistica 13.0 software [59].

## 3. Results

The research was carried out on nine mountain trails leading to the highest peaks of all Beskid mountain ranges located in Poland. The Beskids are the largest group of mountain ranges in Poland, which are part of the Carpathians. They stretch from the Olza to the sources of the San and represent a large natural diversity, including significant differences in landform (e.g., height above sea level), and its anthropogenic development (e.g., distance from the nearest town). Therefore, the trails leading to the highest peaks are characterized by considerable differences in length and difficulty. The shortest trail on which the research was carried out led to Lubomir (3.10 km) in Beskid Makowski, and the longest to Turbacz (15.77 km) in Gorce. For various reasons, including accessibility, tourist development, and, above all, popularity, there were different numbers of tourists on the trails. Figure 2 shows a certain tendency—a large number of tourists is associated with the most popular peaks among tourists. In the summer, Babia Góra (789 people) and Tarnica (391 people) had the most tourists, which is closely related to the fact that, apart from the Tatras, these are the most famous and popular mountain peaks in Poland. It is similar with Turbacz, Radziejowa, and Skrzyczne, where a large number of tourists stay during the summer. The other four peaks were less popular, hence the small difference between the number of people in summer and winter. The big difference between the number of tourists in summer and winter on Babia Góra or Tarnica results from their height, and much lower accessibility, and thus the difficulty (in Babia Góra in winter there is a high avalanche

risk). Although the trails are varied in terms of length, there was no statistically significant correlation between the number of tourists and the length of the trail (Pearson's correlation results for the summer season (r = −0.01, *p* = 0.979) and winter season (r = 0.6, *p* = 0.051)).

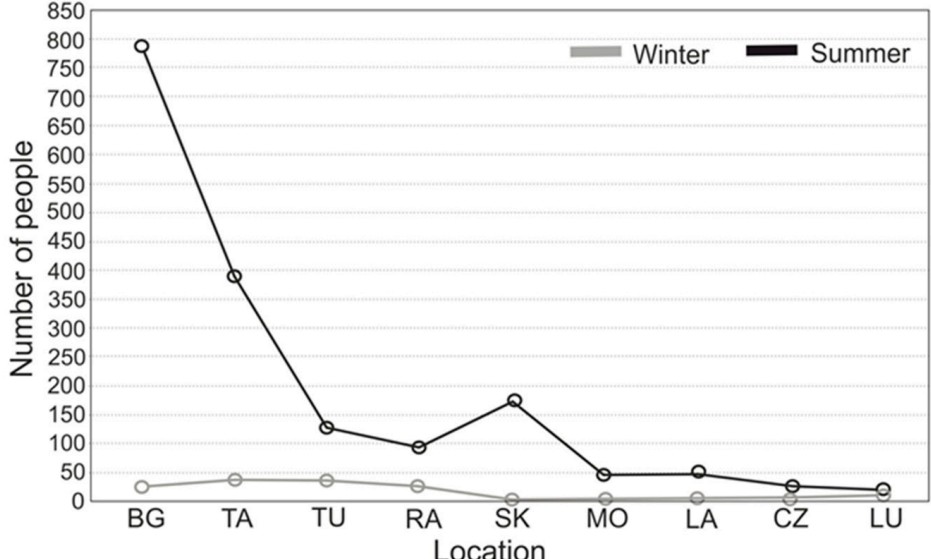

**Figure 2.** Number of persons on the trail during the survey in two seasons (BG—Babia Góra, CZ—Czupel, TA—Tarnica, SK—Skrzyczne, MO—Mogielica, RA—Radziejowa, LU—Lubomir, TU—Turbacz, LA—Lackowa).

The results of the generalised linear model indicated (Figure 2) the differentiation of the average values of sound intensity between the studied locations, while no significant differences in the course of this parameter, with respect to seasons, were found for the same locations (Table 3, Figure 3). In the summer season, Babia Góra, Czupel, as well as Tarnica and Skrzyczne turned out to be the "noisiest" trails. The winter season showed that sound levels were again the highest at Babia Góra. The remaining trails reached a similar average level, with slightly lower values for Radziejowa, Lackowa, and Lubomir (Figure 3).

**Table 3.** Results of the generalised linear model for sound intensity as a function of location (hiking trail/mountain range), season (summer, winter), and multivariate (location × season).

| Effect | St. Sw. | Wald Test | *p* |
|---|---|---|---|
| Free expression | 1 | 154,982.7 | **0.000** |
| Location | 8 | 38.5 | **0.000** |
| Season | 1 | 0.1 | 0.748 |
| Location × season | 8 | 8.7 | 0.369 |

According to World Health Organisation (WHO) guidelines, daytime sound intensity should be in the range of 50 to 55 dB, while night time sound intensity should be 40 to 45 dB [54]. However, these figures apply to urbanised areas and not to naturally valuable natural mountain areas [54–60]. Pilcher et al. [13], investigating the acceptability of sounds by visitors to national parks, reported a value of 37 db (A) as a neutral threshold of acceptability. In the absence of clear guidelines for acceptable sound intensity for natural areas, the lowest value given by the WHO for the night time (40 db) was adopted in this study. The lower values for the night time are based on the need for rest and relaxation, and the authors therefore considered that a value of 40 dB would be most appropriate for areas where it is necessary to reconcile recreational and natural functions. The analysis of the maximum values for both the average sound intensity at individual measurement points and for single second readings showed that the established 40 dB standard was exceeded at all locations and in both seasons. In the case of both average and single data

values, in the summer season, the maximum was reached on the trail to Skrzyczne (average at 68.01 dB, single events at 83.4 dB) and in the winter season on the trail leading to Czupel (average at 68.91 dB, single events at 78.0 db) (Tables 4 and 5). In the case of minimum values, only on one trail to Czupel did the average values exceed 40 dB (reaching 41.11 dB in the summer season). As for the other parameters, they were below the established limit of acceptable sound intensity for tourist areas. The lowest values were recorded in the case of summer, both for average values (35.06 dB) and individual measurements (34.3 dB) on Radziejowa. In winter, on the other hand, the minimum of average values was recorded on Tarnica (36.26 dB), and for single readings on Turbacz (33.1 dB) (Tables 4 and 5).

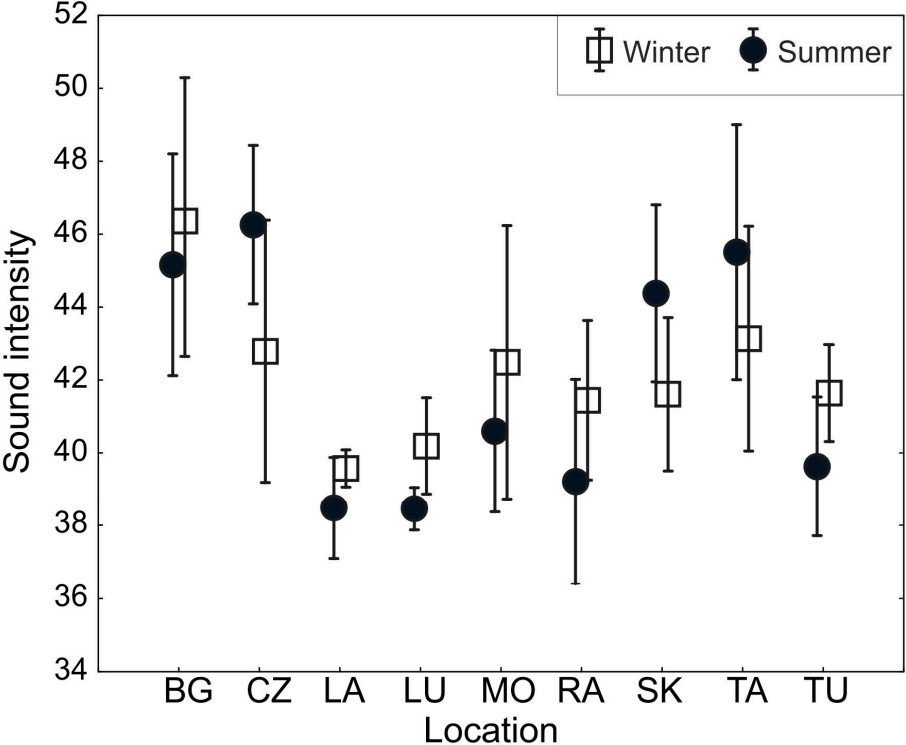

**Figure 3.** Mean ± SE values of sound intensity in the surveyed mountain trails during the summer and winter seasons.

**Table 4.** Summary of maximum and minimum values for average sound levels—summer, winter.

| | Babia Góra | Czupel | Lackowa | Lubomir | Mogielica | Radziejowa | Skrzyczne | Tarnica | Turbacz |
|---|---|---|---|---|---|---|---|---|---|
| **Max summer** | 57.23 | 61.92 | 54.54 | 40.37 | 47.79 | 67.39 | 68.01 | 64.12 | 67.29 |
| **Max winter** | 66.88 | 68.91 | 45.13 | 50.07 | 57.94 | 65.71 | 55.05 | 57.85 | 61.40 |
| **Min summer** | 37.72 | 41.11 | 36.07 | 37.42 | 37.0 | 35.06 | 39.83 | 37.85 | 36.28 |
| **Min winter** | 38.55 | 39.34 | 38.26 | 39.19 | 39.79 | 37.91 | 39.27 | 36.26 | 38.10 |

Analysing the logarithmic averages of sound intensity at individual locations, we could see large differences in the number of measurement points where the 40 dB threshold was not exceeded. In the summer period, the highest number of sites with sound intensity below the standard was on two trails leading to Lackowa (84%) and Turbacz (82%). On the trail leading to Czupel, not a single place with average sound intensity below 40 dB was recorded in the same period. On the trail to Skrzyczne, such places accounted for only 7% (Figure 4).

**Table 5.** Summary of maximum and minimum values of single second readings—summer, winter.

|  | Babia Góra | Czupel | Lackowa | Lubomir | Mogielica | Radziejowa | Skrzyczne | Tarnica | Turbacz |
|---|---|---|---|---|---|---|---|---|---|
| **Max summer** | 67.1 | 76.7 | 66.5 | 52.8 | 56.0 | 78.0 | 83.4 | 71.4 | 76.1 |
| **Max winter** | 75.9 | 78.0 | 59.8 | 61.0 | 69.8 | 71.4 | 67.7 | 64.7 | 77.7 |
| **Min summer** | 37.3 | 39.0 | 35.8 | 37.0 | 36.5 | 34.3 | 38.5 | 37.0 | 36.0 |
| **Min winter** | 38.3 | 39.0 | 38.0 | 38.2 | 38.6 | 37.4 | 39.0 | 35.7 | 33.1 |

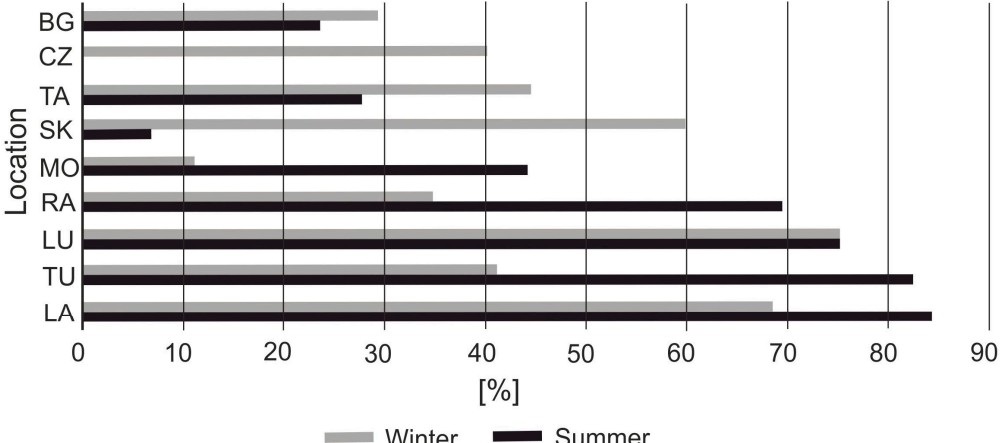

**Figure 4.** Percentage of measurement points where the average sound intensity did not exceed 40 dB (BG—Babia Góra, CZ—Czupel, TA—Tarnica, SK—Skrzyczne, MO—Mogielica, RA—Radziejowa, LU—Lubomir, TU—Turbacz, LA—Lackowa).

During the winter season, no location was recorded where the 40 dB standard was exceeded at all measurement points. The percentage share of points without exceedances ranged from 11% on Mogielica to 75% on the trail to Lubomir.

On the trail leading to Lackowa, the highest peak in the Beskid Niski region, there were few places with an exceedance of 40 dB in both summer and winter. Equally high was Lubomir, which achieved the same result of 75% of readings below 40 dB in both seasons.

The study attempted to create a method to assess the degree of naturalness of the soundscape. To do so, it was necessary to record all sound events and assign them to one of the groups—natural or anthropogenic [35,47,48,50–52].

The GLM analysis showed that in most cases both the type of sound and the season were significantly different (Table 6, Figure 5). In five locations (Turbacz, Lackowa, Lubomir, Mogielica, and Radziejowa), similar relationships were observed, with nature sounds predominating in both seasons, with the total number of recorded sounds being higher in summer.

**Table 6.** Results of the generalised linear model for the valorisation points in the individual mountain trails according to sound type (natural or anthropogenic), season (summer, winter), and multivariate (sound type x season).

|  | St. Sw. | Type | | Season | | Type × Season | |
|---|---|---|---|---|---|---|---|
|  |  | Wald Test | *p* | Wald Test | *p* | Wald Test | *p* |
| Babia Góra |  | 6.20 | **0.013** | 7.20 | **0.007** | 11.60 | **0.001** |
| Czupel | 1 | 4.84 | **0.028** | 4.15 | **0.042** | 14.79 | **0.000** |
| Lackowa | 1 | 28.03 | **0.000** | 2.33 | 0.127 | 0.13 | 0.720 |
| Lubomir | 1 | 8.69 | **0.003** | 1.42 | 0.234 | 0.03 | 0.866 |

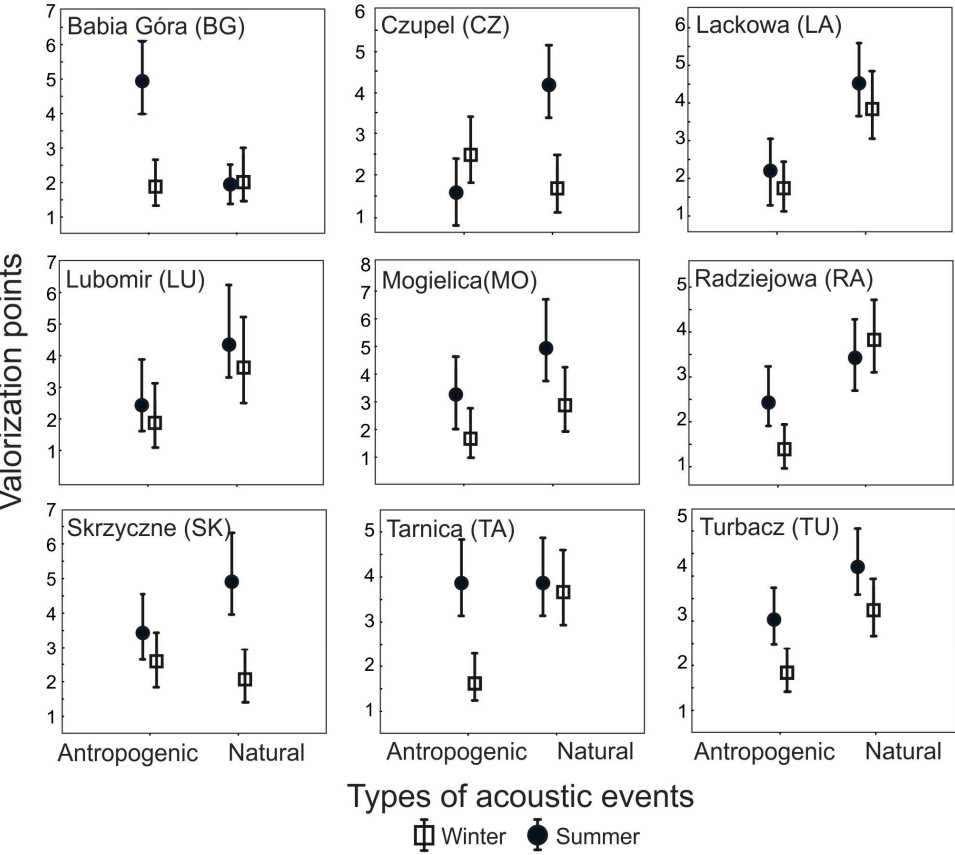

**Figure 5.** Mean ± SE values of natural and anthropogenic value points during summer and winter seasons on individual trails.

The naturalness curves plotted for all trails showed that the location of a particular measurement point had a strong influence on the result. The greatest amount of anthropogenic sounds were usually at the starting points (usually villages) and on the summits or in the area of tourist shelters. A good example is Babia Góra in the summer season—only two points were obtained in location number 1, the car park in Zawoja, and location number 16, the lookout point below the peak; no valorisation points were obtained in location 9, the shelter in Markowe Szczawiny; and one valorisation point was obtained on the very top of the mountain. In the case of only three peaks (Lackowa, Lubomir, and Mogielica) in both seasons was the naturalness curve above three points, which were considered as sites requiring medium or no action. A naturalness curve not falling below three valorisation points was also plotted for the trail to Czupel and Skrzyczne in the summer season, and for Radziejowa in the winter season (Figure 5). The trail to the highest peak of Beskid Żywiecki to Babia Góra was characterised by a rather low naturalness rating of 3.6 points on average in the summer season and 6.2 points in the winter season. The ratings ranged from 0 to 10, with the largest part of the trail being rated between 3 and 7 points. Only in one case in summer and six locations in winter was the maximum number of points (10) reached (Figure 6). The reason for this could be attributed to the high number of tourists on the trail, especially in summer (789 people). The analysis of the recorded acoustic events of anthropogenic type showed that most of them could be classified as socio-cultural (conversations, sneezing, coughing, and emotions of laughter, crying, or shouting) and motoric (noises related to the movement of people, but without the use of combustion or electric propulsion, e.g., steps, sound of trekking poles, creaking of wooden bridges, etc.).

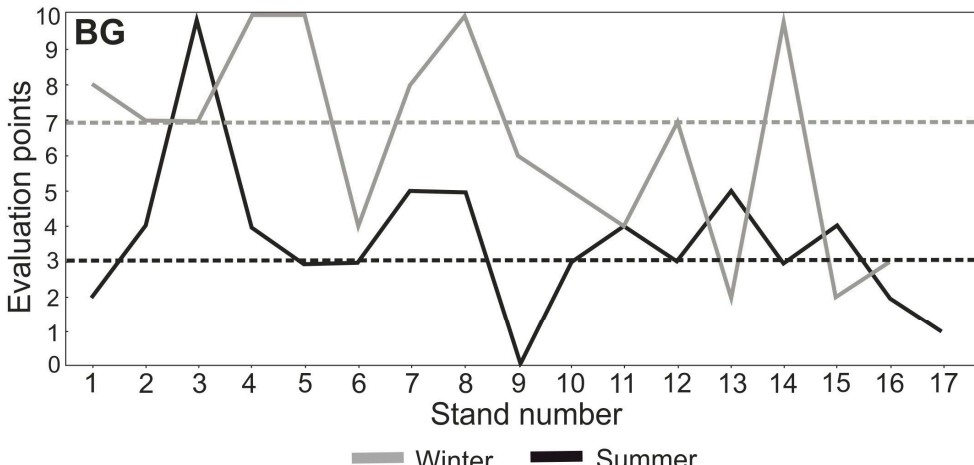

**Figure 6.** Naturalness curves of the soundscape—Babia Góra.

In the case of the Lackowa, Lubomir, and Mogielica trails, similar average scores were obtained in both seasons: Lackowa, summer and winter, with 7.3 points; Lubomir, summer with 6.8 points and winter with 6.9 points; and Mogielica, with 6.8 points in summer and 6.6 points in winter.

The trail to Lackowa was the only one characterised by high naturalness of sound regardless of the season. In summer at 11 points and in winter at 12 out of 19, the soundscape received more than seven valorisation points. In contrast, only one point in winter and two points in summer were awarded the maximum number of points of 10 (Figure 7). Only one point each with the maximum number of points was recorded on the trails to Lubomir and Mogielica (Figures 8 and 9).

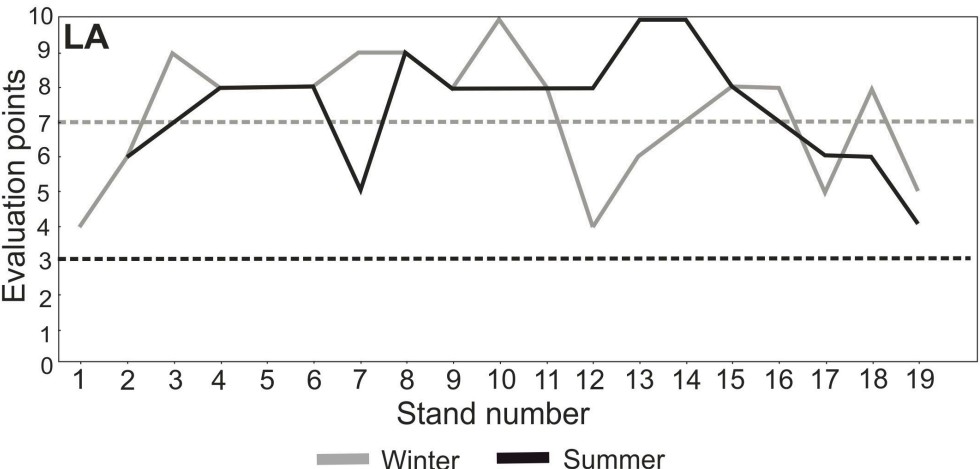

**Figure 7.** Naturalness curves of the soundscape—Lackowa.

In two locations—the trails to Czupel and Skrzyczne—the average number of valorisation points was lower in winter than in summer (Figures 10 and 11). On the trail to Czupel in summer, 11 out of 15 locations were rated above seven points. There were also no sites with very low naturalness below three points. In winter, on the other hand, in three points the naturalness curve was below the limit of three points, and only in two locations it was characterised by high naturalness above seven points. In winter, the average number of valorisation points reached only 4.2, which indicated very low naturalness (Figure 10). In contrast to Babia Góra, most of the anthropogenic sounds belonged to the technical group, mainly the sound of cars, planes, trains, and equipment used in forestry work. Only a few sounds were associated with the direct presence of people, which was related to the small

number of people on the trail—27 people were recorded on the trail in summer and only 2 people in winter.

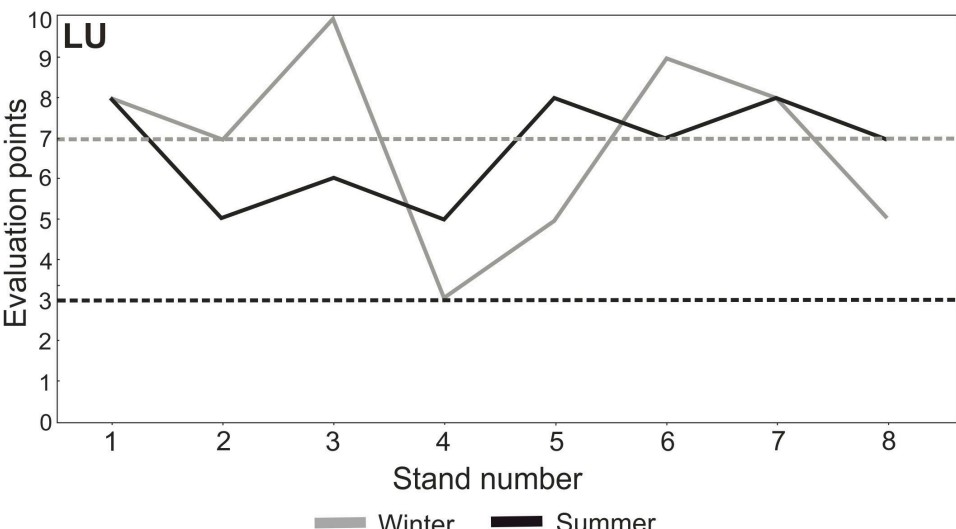

**Figure 8.** Naturalness curves of the soundscape—Lubomir.

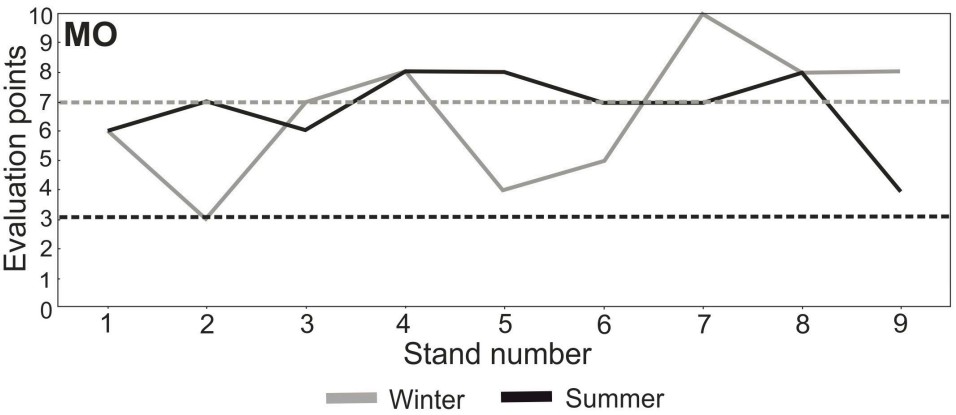

**Figure 9.** Naturalness curves of the soundscape—Mogielica.

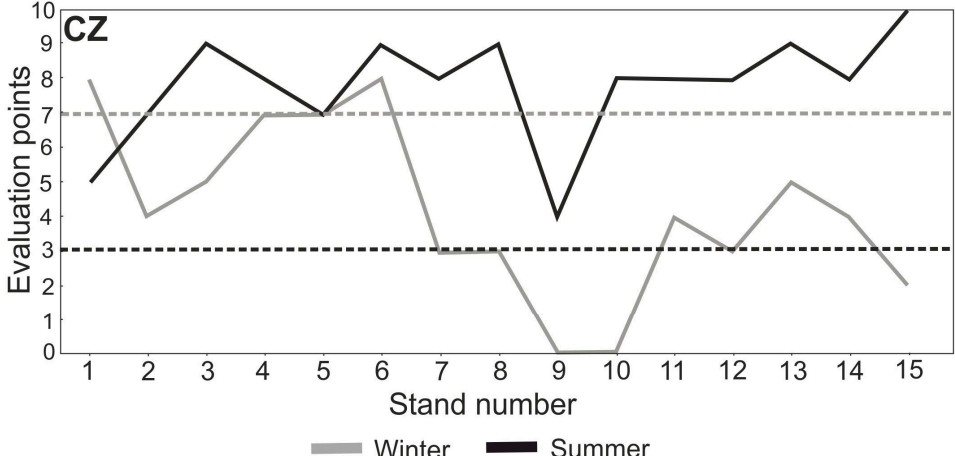

**Figure 10.** The naturalness curves of the soundscape—Czupel.

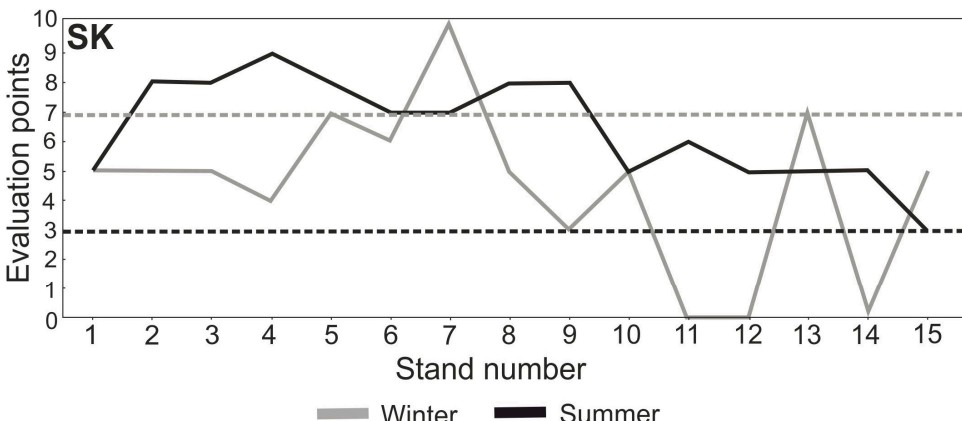

**Figure 11.** Naturalness curves of the soundscape—Skrzyczne.

The trail to Skrzyczne is unique in comparison to the other eight trails; this is due to the presence of the upper station of the chairlift below its summit, which is part of the Central Sports Centre—Olympic Preparations Centre in Szczyrk. Most of the footpaths leading to the highest peak of Beskid Śląski adjoin or even cross the ski slopes or pass under the ski lift lines. Consequently, in winter a large part of the sounds, apart from those of aeroplanes, cars, and forest or agricultural equipment, which dominated at most points, were those associated with the operation of the ski lifts. Among these, the most frequent were the screams and conversations of skiers and snowboarders, the sound of ski lifts and snowmobiles, and the very characteristic sound caused by the friction of skis and especially snowboards on mechanically prepared snow and ice. The average value score in summer was 6.5 points. Not a single place was rated below three points, and as many as 6 out of 15 were above seven points. In winter, the average score was only 4.5 points, in three places the soundscape was assessed as not very natural (below three points), and only one place received a score above seven points (Figure 11).

The trails leading to Radziejowa and Tarnica achieved the highest average number of points in winter—8.0 valorisation points. On Radziejowa, in winter, there was no place rated as very natural (<3 points), while as many as 16 out of 23 sites were rated above seven points, including nine sites characterised by exceptional naturalness (10 points). The trail to Tarnica was also assessed for the most part as very natural (12 out of 18 points) with as many as nine sites receiving 10 points (Figure 12). In the summer season on Radziejowa, the average was slightly lower with 6.5 valorisation points. Most (14 out of 23) places of the trail were assessed as moderately natural, and a fragment of the trails was below three points, while high naturalness was registered in eight points (four places with a score of 10 points) (Figure 12). The naturalness curve in summer on the trail to Tarnica remained mostly in average values; in two locations it fell below three points and at four it was above seven valorisation points. The average number of valorisation points during this period was 5.6 points. (Figure 13).

The soundscape on the trail to the highest peak in the Gorce Mountains (Turbacz) was slightly more natural in the winter season, with an average of 7.3 points, than in the summer, where the average was 6.3 points. In both seasons there were locations rated as not very natural—three in the summer and two locations rated below three points in the winter. In the summer 15 out of 34, and in the winter 18 sites received more than seven points, i.e., sites with a high naturalness of the soundscape. In the winter as many as 12 sites received the maximum number of points, while in the summer there were only three such sites (Figure 14).

During the survey, all people on the trails were also counted. These data provide information on the popularity of a particular trail. In the case of the Skrzyczne trail, only tourists staying directly on the trail in winter were counted; skiers and snowboarders on the

neighbouring ski slope were not included. The biggest differences in the number of visiting tourists concerned four trails (Babia Góra, Tarnica, Turbacz, and Skrzyczne) (Figure 2).

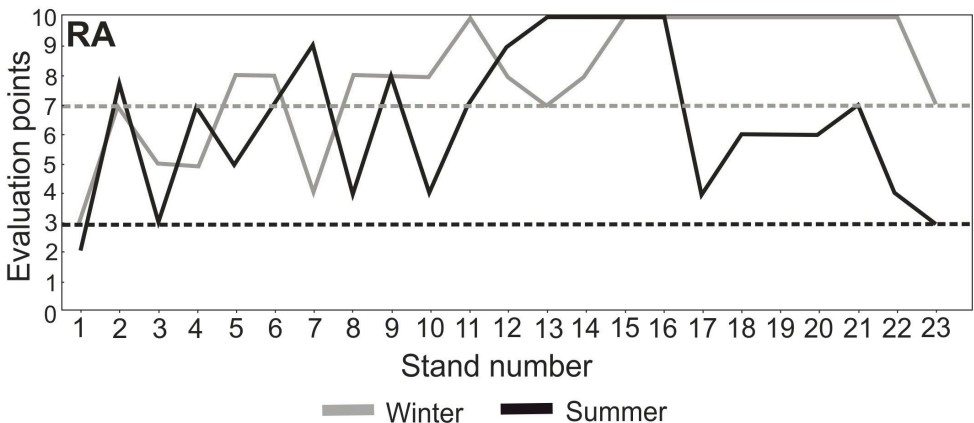

**Figure 12.** Naturalness curves of the soundscape—Radziejowa.

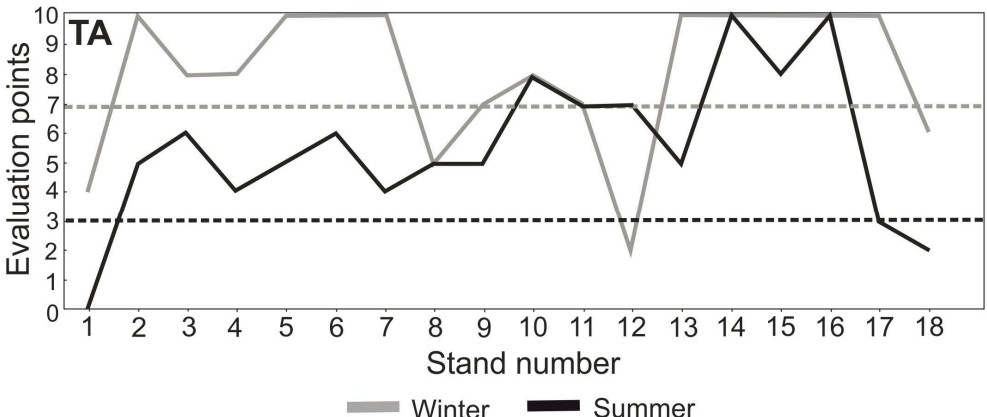

**Figure 13.** Naturalness curves of the soundscape—Tarnica.

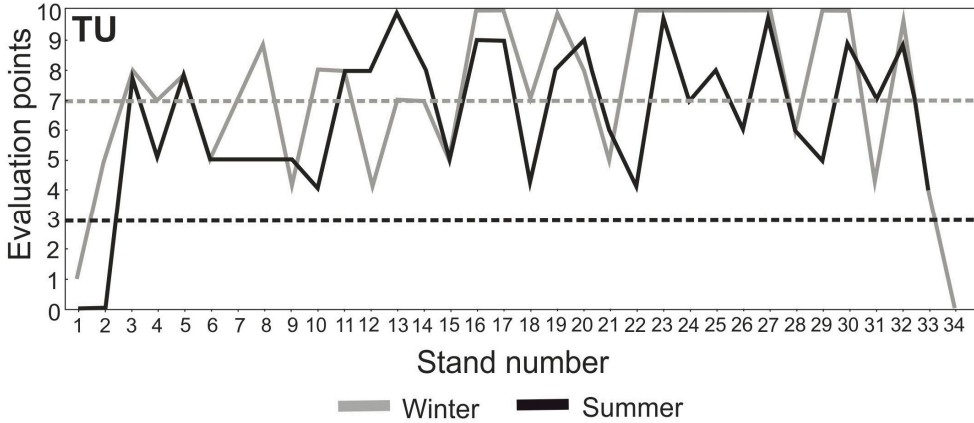

**Figure 14.** Naturalness curves of the soundscape—Turbacz.

## 4. Discussion

　　Today, when considering the acoustic environment, attention is mainly paid to "noise" as a hazard resulting from the physical characteristics of the phenomenon. Because of this, the focus of action is primarily on reducing sound levels to values considered safe for human health (guidelines from the WHO and the European Parliament, among others). According to Hardy [61], measures to reduce street noise resulting from excessive

urbanisation already took place in ancient Rome, and this was strongly correlated with the rapid pace of urbanisation. However, lowering sound levels does not always guarantee sufficient improvement in the quality of the acoustic environment, as the source of sound is an equally important element affecting perception and quality of life [27–31]. An additional problem is the lack of guidelines related to sound intensity in natural areas, especially those used for tourism. While for urbanised areas the values proposed by the World Health Organisation (WHO) as safe for physical and mental health can be used (not to exceed 50–55 dB during daytime and 40–45 dB during night time), they do not work well in recreational areas. The soundscape research methods proposed in ISO 12913 [36–39] mainly concern urbanized areas and the subjective feelings of the recipients. Therefore, the authors proposed a new method based on a simple division of sounds into natural and anthropogenic ones, which increases the method's optimism and allows more attention to be paid to the natural character of sounds without evaluating the feelings associated with them. According to a number of authors [7,10,12,32,33,62] people visiting natural areas such as mountain trails look for peace and quiet; unfortunately, increasingly the sounds and noise they flee from are penetrating these precious areas. This prevents people from fully enjoying the visual beauty of the landscape and fully recovering from the stressful events of everyday life [31]. The residents of areas where these parameters are exceeded are exposed to many adverse changes in both the mental and physical spheres. These include irritability, concentration problems, insomnia, respiratory disorders, blood pressure spikes, increased risk of heart attack, hearing damage, and many others (e.g., [60,61,63–68]).

A varying degree of "sound" anthropopression was found among the nine trails surveyed, and this raises the question of further activities in these areas. On the one hand, quiet and very natural areas could be used to promote the region. After all, tourists are increasingly paying attention to the aesthetics of a space, which is very much linked to its naturalness and beauty [58,69–73]. Work on the suitability and attractiveness of a region for tourism uses methods based on parameters such as the saturation of the area with infrastructure, transport accessibility, the amount of protected areas, accommodation, and catering facilities, or the presence of additional tourist attractions; however, the beauty of the landscape is worth noting in multisensory terms.

As the analysed trails are located in naturally valuable mountain areas, the impact of noise and anthropogenic sound pollution on natural mountain ecosystems is worth noting. Authors such as Wiącek et al. [16], Polak et al. [17], Owens et al. [18], Halfwerk et al. [19], Masayuki et al. [20], Reijnen et al. [21], Mason et al. [22], Goodvin and Shriver [23], Francis et al. [24], and Slabbekoorn et al. [25] primarily highlighted the negative effects of ski area noise on the behaviour and condition of local fauna. Creel et al. [74] and Thiel et al. [75] highlighted increased levels of corticosteroids in the faeces of animals found along ski runs. Noise is also the cause of environmental fragmentation, general ecological stress, and habitat abandonment by wildlife [11,14]. In the study area, in contrast to the ski areas, anthropogenic sound impacts on local biodiversity occur throughout the year, as no significant differences in sound intensity were found between the two study seasons.

Tourism in naturally valuable areas, such as undoubtedly sensitive mountain ecosystems, raises dilemmas related to the conflict between the possibility of developing a local tourism industry and the need to protect nature. Only the development of sustainable tourism can reconcile these two aspects.

## 5. Conclusions

The method of the naturalness curve of the soundscape proposed in this study makes it possible to objectively assess the quality of the landscape without reference to the individual feelings of the viewer, his or her previous experiences, or conditions of cultural origin. As in the case of visual landscapes and their division into natural, primary, or cultural ones, we assess here the saturation with elements of natural and anthropogenic origin. This method therefore becomes universal and can be applied to different areas and cultures without the risk of not taking into account the preferences of the community.

The research showed significant differences in sound levels between the individual hiking trails, while no significant differences were found in the course of this parameter in the two seasons, but in the same locations. The analyses show that on all selected hiking trails and in both seasons, the 40 dB standard proposed by the authors for naturally valuable and recreationally used areas was exceeded. Along the studied trails, there are of course both quiet places, where the sound intensity does not exceed 40 dB, and those where this threshold is significantly exceeded.

When comparing the number of tourists staying on the trails during the measurements, it is difficult to state unequivocally that there is a correlation between sound intensity and naturalness of the soundscape and the number of persons climbing the summit. In the case of, e.g., Babia Góra, which is characterised by both low naturalness and high sound levels, especially in summer, we also have to deal with a large number of tourists using the trail. The recorded anthropogenic sounds are predominantly generated by tourists. However, in the case of the highest peak of Beskid Mały (Czupel), which was the noisiest peak in the summer season, we are dealing with a small number of tourists (27 people), and anthropogenic sounds are mostly generated by means of transport, e.g., aeroplanes, cars, trains, or agricultural tractors. At the same time, the trail was more natural in summer than in winter, even though there were only two persons on the trail during the winter season.

Anthropopressure has had a significant impact on the soundscape of the surveyed trails, but to varying degrees, and no clear influence of tourism as the main factor of disturbance was found. It depends to a large extent on many factors, such as the method of tourist development (e.g., the presence of additional infrastructure in the form of ski lifts or hostels), the popularity of a given trail or the proximity of transport trails, land use and utilisation (e.g., a highly developed settlement network, areas used for agriculture and forestry), and land cover in the form of, e.g., large compact forest complexes or, on the contrary, a mosaic of forest, agricultural, and urbanised areas. At the same time, in the surveyed area of the Polish Beskids, no dependence of the choice of resting place on the quality of the soundscape can be seen. A small number of tourists were also recorded in some quiet and highly natural places, which may indicate other factors determining the choice of a trail by some tourists. This may be evidenced by Babia Góra being characterised by low naturalness and high sound levels and having the highest number of tourists in the summer season at 798 people on the trail at the same time. Taking into account the results of studies by authors such as Manning et al. [9], Merchan et al. [10], Park et al. [8], and Tranel [11] stating that the sounds of nature are of great importance for visitors to naturally valuable areas, the question arises as to whether, in some areas, a certain fashion and popularity of the summit itself is not more important for some tourists than the natural and landscape values.

The study found that there was no clear variation in sound intensity along the studied trails in the same locations but at different times of the year. This, combined with the fact that the naturalness of the soundscape is affected not only by tourist traffic but also by sounds from other sources, means that the pressure on the local ecosystem occurs to a similar degree throughout the year. Of particular concern is that this applies in many cases to areas of national parks or wildlife refuges. This is where the problem of reconciling the needs of tourism and regional development with the protection of naturally valuable areas arises. Promoting such places may lead to an excessive influx of tourists, and the noise they generate may destroy the value that brought them there. In such a situation, it is worth considering whether such areas should be protected rather than promoted to visitors. There is, therefore, a need to continue research into the naturalness of soundscapes, and how they can be protected while developing sustainable tourism in mountain areas.

**Author Contributions:** Conceptualization: M.M.; methodology: M.M.; software: R.K.; validation, M.M.; formal analysis: M.M., R.K. and A.Z.-W.; investigation: M.M.; resources: M.M.; data curation: M.M.; writing—original draft preparation: M.M., R.K. and A.Z.-W.; writing—review and editing: M.M, R.K. and A.Z.-W.; visualization: R.K.; supervision: M.M. and A.Z.-W.; project administration: M.M. All authors have read and agreed to the published version of the manuscript.

**Funding:** This research received no external funding.

**Institutional Review Board Statement:** Not applicable.

**Informed Consent Statement:** Not applicable.

**Data Availability Statement:** Data available on request from the authors.

**Acknowledgments:** The study was conducted with a subsidy of the Ministry of Science and Higher Education in Poland for University of Agriculture in Krakow in 2022. The authors thank unknown reviewers for very valuable comments on a former draft of the manuscript.

**Conflicts of Interest:** The authors declare no conflict of interest. The funders had no role in the design of the study; in the collection, analyses, or interpretation of data; in the writing of the manuscript; or in the decision to publish the results.

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
