# Peer review of "The Method of Soundscape Naturalness Curves in the Evaluation of Mountain Trails of Diversified Anthropopressure—Case Study of Korona Beskidów Polskich"

_sustainability, doi:10.3390/su15010723_

Round 1
Reviewer 1 Report
In this study, the sound intensity value and soundscape naturalness curves were used to assess the soundscape diversity of mountain trails. The paper had sufficient data, novel research methods, scientific analysis and valuable research conclusions. However, there were the following problems, which I hope can be explained and modified.
1. The length of selected mountain trails and the number of measurement points varied greatly, which would cause differences in the analysis results between mountain trails. Please classify whether the results will be caused by the passenger flow load capacity and transportation functions of different mountain trails.
2. Sound intensity and naturalness curves were compared and analyzed respectively. Some sound types may cause excessive sound intensity. Please consider the collaborative analysis between the two and add the superposition analysis of sound intensity on the naturalness curves.
3. There were too many references, but few are published in recent three years. It is suggested to delete the premature literature and supplement the recent research literature.
Author Response
Dear Reviewer,
I appreciate your positive, and, generally, favourable assessment of my article entitled “The method of soundscape naturalness curves in the evaluation of mountain trails of diversified anthropopressure – a case study of Korona Beskidów Polskich”. Thank you for the time and effort spent preparing the review. The comments were very valuable and have helped to improve the quality of the work. We have made every effort to comply with your suggestions and hope that the current version of the work is better than the previous one.
In this study, the sound intensity value, and soundscape naturalness curves were used to assess the soundscape diversity of mountain trails. The paper had sufficient data, novel research methods, a scientific analysis, and valuable research conclusions. However, there were the following problems, which I hope can be explained and modified.
Point 1. The length of selected mountain trails and the number of measurement points varied greatly, which would cause differences in the analysis results between mountain trails. Please classify whether the results will be caused by the passenger flow load capacity and transportation functions of different mountain trails.
Response 1:
Corrected.
The research was carried out on 9 mountain trails leading to the highest peaks of all Beskid mountain ranges located in Poland. The Beskids are the largest group of moun-tain ranges in Poland, which are part of the Carpathians. They stretch from the Olza to the sources of the San and represent a large natural diversity, including significant differences in landform (e.g. height above sea level), and its anthropogenic develop-ment (e.g. distance from the nearest town). Therefore, the trails leading to the highest peaks are characterized by considerable differences in length and difficulty. The short-est trail on which the research was carried out led to Lubomir (3.10 km) in Beskid Ma-kowski, and the longest to Turbacz (15.77 km) in Gorce. For various reasons, including accessibility, tourist development, and, above all, popularity, there are different num-bers of tourists on the trails. The graph (Fig. 2) shows a certain tendency - a large number of tourists is associated with the most popular peaks among tourists. In the summer, Babia Góra (789 people) and Tarnica (391 people) had the most tourists, which is closely related to the fact that, apart from the Tatras, these are the most fa-mous and popular mountain peaks in Poland. It is similar with Turbacz, Radziejowa or Skrzyczne, where a large number of tourists stay during the summer. The other 4 peaks are less popular, hence the small difference between the number of people in summer and winter. The big difference between the number of tourists in summer and winter on Babia Góra or Tarnica results from their height, and much lower accessibility, and thus the difficulty (in Babia Góra in winter there is a high avalanche risk). Although the trails are varied in terms of length, there was no statistically significant correlation between the number of tourists and the length of the trail (Pearson's correlation results for the summer season: r=-0.01, p=0.979, and winter season: r=0.6, p=0.051).
Point 2: Sound intensity and naturalness curves were compared and analyzed respectively. Some sound types may cause excessive sound intensity. Please consider the collaborative analysis between the two and add the superposition analysis of sound intensity on the naturalness curves.
In the case of the examined tourist trails, we are dealing with a very diverse mountain area. This also affects the large differences in length, shape and development on individual routes. Unfortunately, in order to analyze all the mountain ranges belonging to the Beskids in Poland, the authors had to take into account tourist trails of very different lengths. For this reason, the average values were also taken into account in the sound intensity results in order to standardize the results. There is a need for further future research, aimed at establishing a more accurate correlation between the naturalness of sound and the number of tourists, which would help in the management of protected areas (e.g. National Parks) of high natural and tourist value. As suggested, the authors added the results of the analysis of the correlation between the length of the trails and the number of people who were counted during the research to the text. The statistical analysis of the data does not show a significant correlation between the length of trails and the number of visitors.
Response 2:
Corrected.
A very valuable suggestion – the analysis of the superposition of sound waves may be an interesting and valuable supplement to future, and extended research on this area in terms of acoustics. The authors of the work focused on introducing a new method that is a response to the lack of objective methods of soundscape assessment, which in the future may additionally be used to assess changes taking place in environmentally valuable landscapes, and help to manage them properly, especially since these areas are often exposed to excessive environmental pressure caused not only by tourist traffic, but also by the use of these areas by the local community.
Point 3: There were too many references, but few are published in recent three years. It is suggested to delete the premature literature and supplement the recent research literature.
Response 3:
Thank you very much for your valuable suggestion. Where possible, newer literature items have been included in the text. Unfortunately, in some aspects of research, it is difficult to find newer research on this issue. It is especially difficult to find newer literature on the physiological reactions of animals (mainly the study of the concentration of glucocorticoid hormones) exposed to stress caused by human activity related to recreation.
We added new literature references:
Gale, T., Ednie, A., Beeftink, K., 2021. Acceptability and appeal: How visitors’ perceptions of sounds can contribute to shared learning and transdisciplinary protected area governance. Journal of Outdoor Recreation and Tourism, 35, p.100414.
Gale, T., Ednie, A., Beeftink, K. 2021. Thinking outside the park: Connecting visitors’ sound affect in a nature-based tourism setting with perceptions of their urban home and work soundscapes. Sustainability, 13(12), 6572.
Once again thank you for your valuable comments.
Yours sincerely
Magdalena Malec, Renata Kędzior and Agnieszka Ziernicka-Wojtaszek

Reviewer 2 Report
The article presents an investigation of sound pressure levels and its perception in mountain soundscapes. Anthropic and non-anthropic sounds are discussed with reference to noise.
Although the article is well written and structured, there are omissions that should not exist and I ask that it be corrected. It has to do with the validity of the research for other research being done, the universal validity of this article.
ISO 12913 is a reference standard for the study of the characterisation of perception (soundscapes). It talks about questioning different aspects that can then be universally compared (eventful, exciting, pleasant, calm, uneventful, monotonous, unpleasant & chaotic). In the study, perception has been studied with a numerical value, in a single question, which is not well specified. So, firs question is that (A) this aspect should be in the methodology: specify the questionnaire carried out, with content and procedure. There is no reference to this regulation (ISO 12913), so can not been justified with other the studies outside of this one. This way of analysing perception cannot be compared with other benchmarks. Along the same lines, ISO 15666 talks about annoyance, noise; the references to it should take it into account and also this has not been done. Of both ISOs, at least the reference to ISO 12913 is mandatory in these kind of works.
With the aim of furthering these issues, some of the articles cited have this ISO standard as a structure for their work and some of the authors are part of the ISO 12913 working group, which is constantly working on its improvement.
Work such as this article may be of interest for the revision of ISO 12913, why not?, but it must be given an outlet of interest to the scientific community.
In order to specify and give the article an outlet, I think that two issues should be addressed: (B) Justifying why the standard has not been followed; (C) Justifying the interest of the article for this standard. Both issues should be addressed in the introduction, discussion (methodology if necessary) even in the conclusions. It could be a reference to the importance of acoustic levels in landscape regarding with non-anthropic sounds (not covered by the standard because that is a point to work regarding with noise and annoyance which means ISO 15666 and ISO12913 both together). Also it could be the relationship of the question asked in your work (in the perception analysis) to one of the eight questions evaluated by ISO 12913 (eventful, exciting, pleasant, calm, uneven, monotonous, unpleasant & chaotic) and its relationship to sound pressure levels... and, of course, any other aspect of interest to the ISO 12913. In short, the aspects that can be taken into account to evolve the ISO and justify the comparison of this research with others.
Author Response
Dear Reviewer,
I appreciate your positive, and, generally favourable assessment of our article entitled “The method of soundscape naturalness curves in the evaluation of mountain trails of diversified anthropopressure – a case study of Korona Beskidów Polskich”. Thank you for the time and effort spent preparing the review. The comments were very valuable and have helped to improve the quality of the work. We have made every effort to comply with your suggestions and hope that the current version of the work is better than the previous one.
The article presents an investigation of sound pressure levels and its perception in mountain soundscapes. Anthropic and non-anthropic sounds are discussed with reference to noise.
Although the article is well written and structured, there are omissions that should not exist and I ask that it be corrected. It has to do with the validity of the research for other research being done, the universal validity of this article.
Point: ISO 12913 is a reference standard for the study of the characterisation of perception (soundscapes). It talks about questioning different aspects that can then be universally compared (eventful, exciting, pleasant, calm, uneventful, monotonous, unpleasant & chaotic). In the study, perception has been studied with a numerical value, in a single question, which is not well specified. So, firs question is that (A) this aspect should be in the methodology: specify the questionnaire carried out, with content and procedure. There is no reference to this regulation (ISO 12913), so can not been justified with other the studies outside of this one. This way of analysing perception cannot be compared with other benchmarks. Along the same lines, ISO 15666 talks about annoyance, noise; the references to it should take it into account and also this has not been done. Of both ISOs, at least the reference to ISO 12913 is mandatory in these kind of works.
With the aim of furthering these issues, some of the articles cited have this ISO standard as a structure for their work and some of the authors are part of the ISO 12913 working group, which is constantly working on its improvement.
Work such as this article may be of interest for the revision of ISO 12913, why not?, but it must be given an outlet of interest to the scientific community.
In order to specify and give the article an outlet, I think that two issues should be addressed: (B) Justifying why the standard has not been followed; (C) Justifying the interest of the article for this standard. Both issues should be addressed in the introduction, discussion (methodology if necessary) even in the conclusions. It could be a reference to the importance of acoustic levels in landscape regarding with non-anthropic sounds (not covered by the standard because that is a point to work regarding with noise and annoyance which means ISO 15666 and ISO12913 both together). Also it could be the relationship of the question asked in your work (in the perception analysis) to one of the eight questions evaluated by ISO 12913 (eventful, exciting, pleasant, calm, uneven, monotonous, unpleasant & chaotic) and its relationship to sound pressure levels... and, of course, any other aspect of interest to the ISO 12913. In short, the aspects that can be taken into account to evolve the ISO and justify the comparison of this research with others.
Response:
Corrected.
The aesthetic assessment of both the visual and sound landscape, is related to the process of perception, i.e. the conscious, and subconscious reception, but also the comparison of all elements. Due to the fact that these processes take place in the human mind, and are, therefore, associated with many conditions such as age, origin, sex, education, previous experience, emotional state, health and many others, such an assessment is highly subjective. The methods proposed in ISO 12913 are characterized by high subjectivity - the assessment of whether a given sound is pleasant or unpleasant is highly individual. This may mean that the obtained results cannot be compared in different communities or even age groups. Aletta et al. (2019) confirms, that methods A and B proposed in ISO 12913 give similar results, but there are some differences. In addition, he draws attention to the problems of adapting translations, and vocabulary used in different countries, and the resulting divergence of meanings. In addition, taking into account further controversies regarding the proposed methods, and the main purpose of the following studies, the authors decided not to use the above-mentioned methods. The main goal was to answer the frequent accusations against landscape assessment methods about their subjectivity, i.e. an attempt to create an objective method, independent of the recipient's personal feelings. Especially that the research concerns the assessment of specific areas, such as mountain tourist areas visited by groups of tourists very diverse in sociological terms. The authors wanted to assess the soundscape in terms of the participation of natural, and anthropogenic sounds, which would allow to determine the degree of transformation of this landscape, regardless of the personal attitude to individual sounds. More and more authors point to the role of natural sounds in the positive reception of a given area in terms of its tourist values. As suggested in the introduction and discussion, we have added paragraphs on issues from ISO12913. We hope that our current, and future research will help in the development of the standards contained in ISO 12913 and ISO15666.
We added new literature references:
International Organization for Standardization. (2014). ISO 12913-1:2014 Acoustics — Soundscape — Part 1: Definition and conceptual framework. Geneva: ISO.
International Organization for Standardization. (2017). ISO/DIS 12913-2:2017 Acoustics — Soundscape — Part 2: Data col-lection and reporting requirements. Geneva: ISO.
International Organization for Standardization. (2018). ISO/TS 12913-2:2018 Acoustics — Soundscape — Part 2: Data collec-tion and reporting requirements. Geneva: ISO.
International Organization for Standardization. (2018). ISO/WD TS 12913-3 Acoustics — Soundscape — Part 3: Data Analy-sis. Geneve: ISO.
Aletta, F., Guattari, C., Evangelisti, L., Asdrubali, F., Oberman, T., Kang, J. (2019). Exploring the compatibility of “Method A” and “Method B” data collection protocols reported in the ISO/TS 12913-2: 2018 for urban soundscape via a soun-dwalk. Applied Acoustics, 155, pp.190-203.
Papadakis, N. M., Aletta, F., Kang, J., Oberman, T., Mitchell, A., Stavroulakis, G. E. (2022). Translation and cross-cultural ad-aptation methodology for soundscape attributes – A study with independent translation groups from English to Greek. Applied Acoustics, 200, 109031.
Once again thank you for your valuable comments.
Yours sincerely
Magdalena Malec, Renata Kędzior and Agnieszka Ziernicka-Wojtaszek

Round 2
Reviewer 2 Report
Well done... But further research is needed and I propose to go on with it in other papers. Otherwise, it will not be possible to assist in the improvement and evolution of the standard ISO 12913.
You say "The soundscape research methods proposed in ISO 12913 mainly concern urbanized areas, and the subjective feelings of the recipients. Therefore, the authors proposed a new method based on a simple division of sounds into natural and anthropogenic ones, which increases the method's optimism, and allows to pay more attention to the natural character of sounds without evaluating the feelings associated with them." But do not forget that to speak of soundscape is to speak of perception; and to speak of perception is to speak of feelings that are subjective. This characterisation is not at odds with the objective analysis of sound, which is complementary to the study of perception. Both together.